# Peer Support Programs for First Responders: A Critical Review and Research Roadmap

**DOI:** 10.3390/ijerph22101532

**Published:** 2025-10-07

**Authors:** Clint Bowers, Deborah C. Beidel, Victoria L. Steigerwald

**Affiliations:** UCF RESTORES, University of Central Florida, Orlando, FL 32816, USA; deborah.beidel@ucf.edu (D.C.B.); victorialauren@ucf.edu (V.L.S.)

**Keywords:** first responders, peer support, police, firefighters, paramedics

## Abstract

First responders face adverse health effects because they regularly encounter stressful situations and potentially traumatic events. Peer support programs have emerged as a method to reduce these adverse outcomes. A growing interest in peer programs exists despite a restricted body of research in this field. Additionally, the current research on this topic faces significant conceptual and methodological shortcomings. This paper conducts an extensive analysis of present peer support research gaps before proposing future study directions to improve our understanding of this intervention.

## 1. Introduction

The health of first responders is a critical public safety issue [1,2]. These professionals are frequently exposed to stressors such as critical incidents, cumulative trauma, and organizational pressures, which can lead to a range of mental illnesses, including post-traumatic stress disorder (PTSD), depression, anxiety, and substance abuse [3]. Traditional mental health services do not always fully address the unique needs of this population due to factors such as stigma, lack of trust, limited access to culturally competent providers, and a preference for support from those who share similar experiences [4,5].

Peer support programs offer a promising alternative or adjunct to traditional services [6,7]. Peer support programs typically involve training individuals in the workplace to provide emotional, social, and practical support to their colleagues. Since these programs typically provide support at no cost and are delivered flexibly so that support can be offered on duty or at other times convenient for first responders, they reduce the impact of several barriers to care often reported by first responders [8]. However, the scientific evidence base for peer support interventions, particularly among first responders, has several limitations [9,10,11]. We describe these limitations and provide suggestions for improvement below.

## 2. Research Challenges in Peer Support

### 2.1. Conceptual/Theoretical Problems

Research on peer support faces challenges due to conceptual ambiguities, which hinder our understanding of the mechanisms underlying intervention effects and the optimal design and implementation of peer support programs. These include the following:

#### 2.1.1. Construct Definition

There is currently no prevailing definition of “peer support.” The term is used broadly, describing everything from informal, ad hoc interactions to highly structured, formal programs. This lack of an accepted definition makes it challenging to compare findings across studies and synthesize results [10,11]. Additionally, the definition of a ‘peer supporter’ or ‘peer support training’ is ambiguous and ranges from one-day workshops to certification as a ‘recovery peer specialist’ after completion of 40 h of class training and 500 h of supervised work experience (https://flcertificationboard.org/certifications/certified-recovery-peer-specialist-crps// (accessed on 1 October 2025)). Recent guidelines for peer support, such as those published by the Mental Health Commission of Canada (https://mentalhealthcommission.ca/wp-content/uploads/2021/09/Guidelines-for-the-Practice-and-Training-of-Peer-Support.pdf (accessed on 1 October 2025)) and the Atlas Institute (https://atlasveterans.ca/documents/peer-support-guidelines-en.pdf (accessed on 1 October 2025)), address some of these definitional ambiguities. A recent article proposing a typology of peer support [12]. They include three types of peer support: peer-led, peer-enabled, and peer partnership. The authors also distinguish between program and service delivery, yielding six categories.

Finally, there are differences in the purpose of the peer support program [13]. For example, some programs adopt a primary prevention approach, which includes surveillance and referral that is not tied to a specific traumatic event (e.g., UCF REACT). Others focus on reacting after a trauma has occurred, such as CISM. Although both are peer support programs, they have vastly different goals and activities.

Recommendations: Although there is no accepted definition of peer support, common themes in existing studies emphasize that peer support is voluntary, involves interactions between people with shared experience, and is targeted at supporting colleagues in the workplace. At a minimum, studies in this area should describe whether these three criteria are met. Adopting the typology from Price and her colleagues might provide additional clarity.

In an effort to develop peer support guidelines grounded in evidence, Creamer and colleagues [6] reached a consensus that peers are members of the population served who receive initial and ongoing training in skills that enable them to effectively support others (e.g., active listening). In future research, it would be beneficial if researchers would describe the training that peer supporters have received. Ideally, scientists in this area will agree on a set of minimum requirements to be recognized as legitimate “peer support training”. The guidelines provided by the International Association of Chiefs of Police might serve as a useful model for these requirements. Failing that, researchers should provide sufficient detail about the content, duration, and frequency of training to allow readers to infer the knowledge and skills that the supporters were expected to possess.

#### 2.1.2. Absence of Explicit Theoretical Frameworks

Many studies fail to articulate a theoretical framework. Atheoretical research does not articulate underlying mechanisms of action that generate testable hypotheses. Indeed, hypotheses are often not even stated in research publications. While this may be less problematic in qualitative research, it limits the utility of quantitative studies whose objectives are to test explicit hypotheses.

Recommendation: Editors and reviewers should require an explanation of the underlying theoretical basis and clear articulation of the proposed mechanism of action. Furthermore, the hypotheses that follow from the model would be articulated and specifically tested. Candidate theories include social support theory [13], social cognition theory [14], and post-traumatic growth theory [15]. For instance, if peer support is designed to address the trauma of critical incidents, then outcomes related to PTSD may be relevant. However, if the goal is to provide nonjudgmental support or link to professional resources, then outcomes might be different, related to perceived support (as per our recommendation regarding social network analysis) or access to EAP/counseling.

#### 2.1.3. Overlooking Contextual Factors

Peer support research often overlooks the crucial role of organizational context, leadership, and diversity factors that may influence the success of interventions. When the ‘organizational context’ is not adequately characterized, it is challenging to evaluate how different environments affect the program’s initial conditions [16]. Failing to address these contextual variables sufficiently means that the findings may not be transferable and also limits the ability to accurately interpret the findings.

Recommendations: Researchers in this area should provide as much information as possible to describe contextual factors that influence the delivery or effectiveness of these programs. Damschroder and her colleagues [16] articulate how the Consolidated Framework for Implementation Research (CFIR) can be used to ensure all important contextual factors are included. Examples include:Participant Characteristics: These encompass the attributes of both peer supporters and the individuals receiving support. For peer supporters, this might include their experience with mental health challenges, their motivation to help others, their communication skills, or their pre-existing knowledge of mental health resources. It might also include the method by which they became a peer specialist. Did they self-nominate? Were they selected based on specific characteristics? For those receiving support, relevant characteristics may include their current mental health status, willingness to engage in peer support, demographic information, and career details.Program Resources: These include tangible assets (e.g., funding, meeting space, personnel to lead or coordinate program) and intangible assets allocated to the peer support program, such as leadership buy-in. It would also be beneficial to specify whether a program is manualized or uses an existing curriculum.

Organizational Climate: Key variables in the organizational climate include trust (between peers and within the organization), perceptions of confidentiality, leadership behavior, safety culture, and managerial support for mental health [16,17].

### 2.2. Methodological Issues

Several methodological issues limit the usefulness of the current research on peer support. These are described below:

#### 2.2.1. Research Design Limitations

Many studies in peer support research employ qualitative or descriptive designs. There is a shortage of randomized controlled trials (RCTs) or other robust quantitative methodologies [10,11]. Though randomization to groups may not capture the process of voluntarily deciding whether to participate in a peer support program [16,18], and RCTs can be very difficult to implement in organizational research due to challenges in controlling potentially confounding variables, the reliance on less rigorous designs makes it difficult to establish causal relationships between peer support interventions and desired outcomes. It is also challenging to precisely identify which specific inputs (e.g., type of training, characteristics of peer supporters) are most critical for program success. Furthermore, the effectiveness of different peer support processes cannot be definitively compared or optimized. Researchers cannot control potentially confounding factors, making it difficult to make causal claims. Finally, the prevalence of small sample sizes and the use of convenience sampling limit the external validity of findings from these studies. However, it is important to note that high-quality research can be produced using qualitative approaches without including hypotheses, and qualitative studies can provide valuable insights into context and process. Combining qualitative and quantitative techniques in multi-method designs can also enable researchers to examine context and test hypotheses.

Recommendations: Researchers should design and implement more robust research designs to study peer support. These include:

Longitudinal Studies: Implement studies with multiple assessment points (e.g., pre-intervention, post-intervention, 3-month, 6-month, 12-month follow-ups) to capture both immediate and sustained effects.

Randomized Controlled Trials (RCTs) or Quasi-Experimental Designs: Where feasible, utilize RCTs to establish causality. When randomization is not possible, employ rigorous quasi-experimental designs with matched control groups to minimize confounding variables.

Multi-Site Studies: Conduct research across multiple first responder agencies and diverse geographical locations to enhance the generalizability of the findings.

#### 2.2.2. Assessment Tool Inadequacies

A significant methodological weakness arises from the lack of standardized, validated outcome measures. Most studies rely on self-report instruments, which may be subject to bias and have limitations in capturing behavioral outcomes, though they remain useful for constructs such as perceived support, coping, and stigma. The collection of quantitative data often occurs without proper statistical analysis and without reporting effect sizes or determining significance levels. Finally, it is essential to measure outcomes that align with the program’s goals. For example, a primary prevention-oriented program’s impact might be better assessed by the number of referrals, while a crisis-response program could be assessed with PTSD diagnoses after an event.

Recommendations: Researchers should utilize standardized measures to evaluate peer support outcomes at all relevant levels. Examples include:Individual Outcomes: Employ a battery of validated, standardized clinical and psychological assessments to measure changes in mental health symptoms (e.g., PHQ-9 for depression, GAD-7 for anxiety, PCL-5 for PTSD), coping skills, and resilience. Use self-report measures for help-seeking behaviors and perceived social support. Incorporate objective measures where possible, such as physiological markers of stress or behavioral indicators from simulated scenarios.Interpersonal Outcomes: Utilize social network analysis to map changes in support networks. Administer surveys assessing communication patterns, perceived social support, and perceived stigma within peer groups.Organizational Outcomes: Use archival data on absenteeism, turnover rates, and workers’ compensation claims. Organizational climate surveys should be administered at different times to measure the changes that result from peer support.

#### 2.2.3. Sampling Challenges

The research methods used in peer support studies typically recruit participants from either a single organization or a restricted geographic area. Demographic information is either incomplete or completely absent in many cases. The research fails to address diversity aspects, including gender differences, ethnic backgrounds, and professional roles. While studies may employ various non-random sampling techniques, such as purposive sampling, quota sampling, or snowball sampling, the reliance on these methods can limit the external validity of the findings.

Recommendations: Researchers in this area should endeavor to use larger, more representative samples. When possible, multiple regions and organizations should be included. Researchers can also employ various matching strategies to better reflect the underlying population. Researchers should recruit and retain participants from underrepresented groups (e.g., women, racial and ethnic minorities, diverse professional roles) to ensure the research is inclusive and applicable to the entire first responder community. Finally, authors should provide an estimate of statistical power for all results.

#### 2.2.4. Failure to Measure Critical Processes

Ultimately, the effectiveness of peer support is dependent on the behaviors that occur between coworkers. Unfortunately, this is a challenging area to research. It is, however, crucial that researchers undertake this work to enhance our understanding of this intervention. The aforementioned Atlas guidelines review many of the issues that should be considered. Examples include:Peer Interactions: The essential element of peer support is the interaction between peer supporters and their colleagues. As highlighted by the Atlas Recommendations, peer support interactions include the number of contacts, their duration, and the communication modality (one-on-one conversations, group meetings, or online forums).Program Implementation Fidelity: This refers to the extent to which the peer support program is delivered as intended. It refers to the degree to which supporters adhere to established protocols.Supervision and Ongoing Training: The training and development of peer supporters are crucial to the success of these programs. This cluster includes activities like regular supervision sessions led by mental health professionals, debriefing opportunities, and ongoing training.Referral Process: Peer supporters should be trained in available community resources and the processes for engaging them.

Recommendations: Researchers should implement observational studies (e.g., coding of recorded interactions) to assess the quality and content of peer support interactions. They can also utilize self-report measures from both peer supporters and recipients regarding perceived helpfulness and interaction characteristics. Process tracing and social network methods can be employed to understand the flow of support [14]. Additionally, the program should conduct routine audits of its activities and training delivery to verify both effectiveness and quality. Qualitative information about challenges related to adherence can be obtained through interviews with trainers and supervisors. To assess whether the goal of connecting peer support recipients to appropriate resources is being met, programs can evaluate peer supporters’ familiarity with available resources and track and report on the number and type of referrals they provide. Finally, scientists should track participation in supervision and ongoing training sessions. Evaluate the content and perceived utility of these activities through surveys and qualitative interviews with peer supporters.

#### 2.2.5. Failure to Measure Critical Outcomes

The effects of a peer support program should be observed across multiple levels, including individual, interpersonal, and organizational levels. Evaluating outcomes at each of these levels provides a comprehensive understanding of the program’s overall impact. These include:

Individual Outcomes: These focus on changes within the individuals who received peer support. They might include improvements in mental health symptoms (e.g., reduced anxiety, depression, PTSD symptoms), increased coping skills, improved occupational functioning, enhanced resilience, reduced feelings of isolation, and increased help-seeking behaviors [19].

Interpersonal Outcomes: These focus on changes in relationships and social behavior. Possible peer support outcomes include improved communication among peer groups, enhanced social support networks, or a reduction in the stigma surrounding mental illness.

Organizational Outcomes: These refer to the broader effects on the organization. Examples of organizational outcomes include improvements in the organizational climate, increased morale, reduced absenteeism, or lower turnover rates [20].

Recommendations: To accurately describe the condition of participants, researchers should utilize standardized psychological assessments to understand the characteristics of peer supporters (e.g., communication skills, empathy, resilience). We should also collect and report detailed demographic and professional background information about peer supporters and those they support. A standardized assessment of the frequency and severity of potentially traumatic events should also be included. Surveys and interviews with program administrators should be used to assess perceived resource adequacy and utilization. Document analysis of program budgets and training materials can also provide objective data. Finally, scientists should utilize validated organizational climate surveys to assess perceptions of mental health support, leadership buy-in, and stigma. Conduct interviews with leadership and frontline personnel. Analyze existing organizational policies related to mental health and well-being [21].

#### 2.2.6. Support for Peer Providers and Organizational Optimization

Research should also investigate the support required for peer support providers themselves, given the potential for burnout or low quality support. This includes examining organizational-level supports to optimize the adoption and reach of peer support services.

## 3. Conclusions

The current state of peer support research for first responders reveals a field with substantial promise but significant methodological and conceptual limitations that impede meaningful progress. This comprehensive analysis has identified eight critical areas requiring attention: the absence of standardized definitions and theoretical frameworks, inadequate research designs, insufficient assessment tools, sampling challenges, and failure to measure essential processes and outcomes across individual, interpersonal, and organizational levels.

The conceptual ambiguities surrounding peer support interventions represent perhaps the most fundamental barrier to advancing this field. Without consensus on what constitutes peer support, peer supporter qualifications, or minimum training requirements, researchers cannot build upon each other’s work. Furthermore, practitioners cannot develop evidence-based programs with confidence. The lack of explicit theoretical frameworks further compounds this problem by preventing the development of testable hypotheses and a mechanistic understanding of how peer support interventions produce their effects.

Methodologically, the field’s overreliance on qualitative and descriptive designs, combined with inadequate sample sizes and limited geographic diversity, severely constrains the generalizability and causal interpretation of findings. The absence of standardized, validated outcome measures across individual, interpersonal, and organizational domains means that even well-designed studies may fail to capture the full impact of peer support interventions. These limitations are particularly problematic given the complex, multi-level nature of peer support programs and their intended effects.

The recommendations presented in this analysis provide a roadmap for transforming peer support research from its current state of limited utility to a robust evidence base that can inform practice and policy. The call for standardized definitions, explicit theoretical frameworks, rigorous experimental designs, validated assessment tools, and comprehensive process evaluation represents a necessary shift toward scientific rigor. Equally important is the emphasis on measuring outcomes across multiple levels of analysis and including diverse populations to ensure that research findings are externally valid.

The stakes for improving peer support research extend far beyond academic considerations. First responders continue to experience elevated rates of mental health problems, and peer support programs represent one of the most widely implemented yet poorly understood interventions in this population. The failure to establish a strong evidence base not only perpetuates uncertainty about program effectiveness but also risks the continuation of ineffective practices and the missed opportunity to optimize interventions that could significantly improve first responder health.

## 4. Recommendations for Future Research

Moving forward, the research community must prioritize collaborative efforts to address these foundational issues. This includes developing consensus definitions, establishing minimum standards for peer supporter training, creating validated assessment batteries, and conducting large-scale, multi-site studies using rigorous experimental designs. This will likely require consensus among researchers, practitioners, and stakeholders. In particular, partnerships between first responder agencies and researchers could facilitate rigorous study of peer support program development and implementation. The ultimate goal is not merely to improve research quality but to ensure that first responders have access to peer support programs that are demonstrably effective in reducing adverse mental health outcomes and promoting resilience. Achieving this goal requires immediate action on the recommendations outlined in this analysis, sustained commitment from researchers and funding agencies, and recognition that methodological rigor is essential for translating good intentions into meaningful outcomes for those who serve our communities.

## Data Availability

Data sharing is not applicable.

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
