# Peer review of "Peer Support Programs for First Responders: A Critical Review and Research Roadmap"

_ijerph, 2025, doi:10.3390/ijerph22101532_

Round 1

Reviewer 1 Report

Comments and Suggestions for Authors

The authors raise important issues related to gaps in research on peer support in the public safety community.  There are indeed research challenges that need to be addressed, but my main concern is that they are missing key literature that has already been published in this field.  If making a case for gaps, need to anchor the ideas in what is already known.  There are, for example, best practice guidelines regarding peer support published by the Mental Health Commission of Canada https://mentalhealthcommission.ca/wp-content/uploads/2021/09/Guidelines-for-the-Practice-and-Training-of-Peer-Support.pdf, and recent guidelines for peer support specifically for First Responders published by the Atlas Institute: https://atlasveterans.ca/knowledge-hub/peer-support/peer-support-guidelines-for-veterans-military-public-safety-personnel-and-their-families/   Both of these documents address some of the gaps that you have noted. 

Initial context is needed regarding current models of peer support in the field (e.g., CISM, CISD) since there is a body of literature looking at the efficacy/effectiveness of these approaches.  There are also several review papers missing (e.g., Anderson et al., 2020; Beshai & Carleton, 2016) - look for "public safety personnel" as a key search term in addition to 'first responders'.  Also, there is a recent article proposing a typology of peer support which can add to your reflections on construct definitions (Price et al., 2022).  

When discussing research design, it is important to consider implementation science approaches which examine the influence of contextual factors (culture, resources) and the influence of the implementation process on outcomes (e.g., CFIR framework).  These approaches are grounded in 'real world' implementation and may provide more applicable findings than traditional RCT's. 

Suggest that you make more explicit links between theoretical approaches and proposed outcomes.  There should be congruence, for example, between the proposed mechanisms of action and impact evaluation (e.g., if peer support is designed to address the trauma of critical incidents, then outcomes related to PTSD may be relevant, but if the goal is to provide nonjudgemental support or link to professional resources, then outcomes might be different related to perceived support (as per your recommendation re: social network analysis) or access to EAP/counselling). 

Need to critically reflect on what the realistic outcomes might be for a peer support encounter - what might be the dose/response required to effect change at an organizational level?  I agree that we need to gather more foundational information about the frequency, duration and type of peer support encounters.  One of the challenges can be resistance to documentation of what peers consider to be a private, anonymous, potentially informal connection which can be a barrier to formalized research. 

Some comments were unclear (e.g. lines 204-206 regarding the need to utilize standardized psychological assessments to understand characteristics of peer supporters).  Are the right questions being asked?   I wonder if  there is value, for example, in recommending research looking at the supports required for the peer support providers (given the potential for burnout or low quality support)?  Do we need to research on supports required at an organizational level to optimize the adoption and reach of the peer support service? 

I don't disagree that a critical review and research roadmap is needed, but paper could be strengthened with clearer grounding in the literature and critical reflections on optimal research design.

Author Response

We appreciate the reviewer's insightful comments and constructive feedback, which
have significantly strengthened our manuscript. We have carefully considered each
point and revised the manuscript accordingly. Our responses to the specific concerns
are detailed below.
Concern 1 : Missing Key Literature and Lack of Anchoring Ideas in Existing
Knowledge
The reviewer correctly points out the importance of anchoring our discussion of
research gaps in existing literature, particularly regarding best practice guidelines and
current models of peer support. We acknowledge that our initial submission did not
adequately incorporate some key resources.
Response: We have thoroughly reviewed and integrated the suggested literature into
the revised manuscript. Specifically:
We have incorporated references to the best practice guidelines published by the
Mental Health Commission of Canada and the Atlas Institute. These documents are now cited in lines 48-54 to provide a more comprehensive overview of existing frameworks and to contextualize our discussion of definitional ambiguities.

We have added a brief discussion of current models of peer support in the field, such as Critical Incident Stress Management (CISM) in the Introduction [lines 60-64]. This provides essential initial context and acknowledges the existing body of literature on these approaches.

We have included the suggested review papers (e.g., Anderson et al. ; Beshai & Carleton, ) and expanded our search terms to include'public safety personnel' to ensure a more comprehensive literature review. These additions enhance the foundational understanding of the field [Line 45 and associated references].

The recent article proposing a typology of peer support (Price et al., ) has been integrated into the introduction to enrich the discussion on construct definitions and provide a more nuanced perspective on the varied forms of peer support [Lines 55-58 & reference].

Concern 2 : Consideration of Implementation Science Approaches
The reviewer suggests incorporating implementation science approaches, particularly with regard to contextual factors and the impact of the implementation process on outcomes.

Response: We agree that implementation science offers valuable frameworks for understanding the complexities of real-world interventions. We have revised the manuscript to include a discussion of implementation science approaches, such as the CFIR framework. We emphasize that these approaches can provide more applicable findings than traditional RCTs by examining the influence of contextual factors (culture, resources) and the implementation process on outcomes. [Line 40, 202, 357-359].

Concern 3 : Explicit Links Between Theoretical Approaches and Proposed Outcomes
The reviewer highlights the need for more explicit links between theoretical approaches and proposed outcomes, emphasizing congruence between mechanisms of action and impact evaluation.
Response: We have revised Section entitled "Absence of Explicit Theoretical Frameworks" to more clearly articulate the congruence between theoretical approaches and proposed outcomes. We now provide examples of how specific theoretical frameworks (e.g., social support theory, post-traumatic growth theory) can inform the selection of relevant outcomes. For instance, if peer support aims to address trauma, PTSD-related outcomes are relevant; if the goal is non-judgmental support or linking to professional resources, outcomes related to perceived social support or access to EAP/counseling would be more appropriate. This revision ensures a more coherent and testable framework for future research [Lines 89-95].

Concern 4: Critical Reflection on Realistic Outcomes and Supports for Peer Providers
The reviewer raises important questions about the realistic outcomes of peer support encounters, the dose/response required for organizational change, and the need to research supports for peer support providers.
Response: We acknowledge the reviewer's astute observations regarding the realistic outcomes of peer support and the challenges associated with documenting informal connections. We have expanded Section "Failure to Measure Critical Outcomes" to include a more critical reflection on what constitutes realistic outcomes for peer support encounters, considering the potential for informal and private interactions. We also emphasize the need for research to gather more foundational information about the frequency, duration, and type of peer support encounters, while also recognizing the inherent challenges in formalizing such data collection. Furthermore, we have incorporated the reviewer's excellent suggestion to research the supports required for peer support providers. [Lines 256-260]

Concern 5: Unclear Comments and Asking the Right Questions
The reviewer found some comments unclear regarding the need to utilize standardized psychological assessments to understand characteristics of peer supporters and questioned whether the right questions were being asked.
Response: We have clarified the text to emphasize that standardized psychological assessments
should be used to understand the characteristics of peer supporters, not necessarily their mental health status in a diagnostic sense. The intent is to gather baseline data on attributes that might influence their effectiveness as peer supporters (e.g.,
communication skills, empathy, resilience). We have rephrased this section to ensure clarity and avoid misinterpretation.
[Lines 244-247].

Concern 6: Paper Could Be Strengthened with Clearer Grounding in Literature and Critical Reflections on Optimal Research Design
This overarching concern summarizes several points made by the reviewer.
Response: We believe that the revisions made in response to the preceding concerns
directly address this overarching point. By incorporating key missing literature,
explicitly linking theoretical approaches to outcomes, and expanding our discussion
on research design to include implementation science, we have significantly
strengthened the manuscript's grounding in existing knowledge. Furthermore, our
enhanced critical reflections on optimal research design, including the nuanced
discussion of qualitative and quantitative methodologies (as also suggested by
Reviewer ), contribute to a more robust and well-rounded paper. We are confident
that these changes have made the manuscript more comprehensive, rigorous, and
impactful.

Reviewer 2 Report

Comments and Suggestions for Authors

This manuscript presents a review and set of recommendations concerning peer support programs designed to improve the mental health of first responders. This topic is important to examine given the high number of work-related mental health problems that first responders experience. It is also a topic worth examining given the increasing volume of peer-reviewed and popular literature on the subject. I like that the authors present a critique of the current literature and provide constructive, specific, and achievable  recommendations for each of the criticisms they raise. In each case the recommendations provided can be easily linked to the criticism of the literature.

I think the paper makes a needed contribution to the discussion around peer support mental health programs. However, I think there are some things in the paper that should be addressed before it is accepted at Environmental Research and Public Health.  I do not see the paper requiring major changes but I believe the changes that I am suggesting are more than just minor editorial revisions.  Addressing these things will make for a stronger and more influential paper.  Therefore, my overall recommendation falls between “Accept after minor revision” and “Reconsider after major revision.”  

The paper provides a fulsome critique of the literature, noting the lack of conceptualization and operationalization of key concepts (including peer support), the under-theorizing of study approaches, inadequate descriptions of relevant contextual factors, and a general failure to adopt a set of standardized and validated measures which would enable study-to-study comparisons. Another positive in the paper is that research guided by the recommendations will have real world impact. That is, the criticisms of the research literature and how they will be addressed will facilitate the uptake and adoption of the programs in organizations, making them more effective and efficient, and enabling practitioners and leaders to gather more reliable data that they can act on.

The writing in the paper is very good and generally free from errors. The sections are well-organized and they flow logically.  The use of critique + recommendation structure that appears in each section is well done and intuitive.

There are spots throughout the manuscript where I think the authors need be more balanced in their criticism of studies that do not use RCT or other quantitative study designs. The reality of organizational research or research that takes place within an organization is that RCTs and even quasi-experimental designs are very difficult as it so challenging to control potentially confounding variables.  For example, in the section (2.1.2) on theoretical frameworks, this statement appears, “Indeed, hypotheses are often not even stated in research publications.” If the goal is not to test hypotheses, such as in most qualitative studies, why would hypotheses be stated? It is certainly a problem if the objective of the study is to test hypotheses and the criticism is warranted.  I would also note that while hypothesis testing can produce excellent research, it is not synonymous with high-quality research. A researcher can produce high-quality research using qualitative approaches without including hypotheses. In this section and in other spots the manuscript would be improved if the critiques were a little more sophisticated and nuanced. I also think the authors could at certain spots note the advantages of employing qualitative study designs or at least techniques and even combining these with quantitative techniques in multi-method designs. Doing so, can enable researchers to examine context and also test hypotheses.   

The following are specific comments.

In the paper’s opening paragraph, line 19, I would add “such as police officers, firefighters, and paramedics”

Section 2.1.3, the authors note that much of the research glosses over the organizational context which limits the researcher(s) abilities to interpret and explain their findings. On line 77-78, They specifically note “Failing to address these contextual variables sufficiently means that the findings may not be transferable.” Concern about the generalizability of findings is important and warranted but I would argue that having the information about context is also critical for interpreting the findings themselves.

There are also spots in the manuscript where I think the authors need to add a few evidentiary examples to support their assertions. In many of these instances, I am reasonably confident the views expressed by the authors are accurate but without evidence to support the statements, the readers cannot be sure. For example, line 65 states, “Many studies fail to articulate a theoretical framework.” Reference to 2-3 studies where no theoretical framework is explicitly identified should be included here.  A similar situation emerges regarding statements about convenience sampling. See point below.

In section 2.2.1, line 111-112, it is noted that the use of “convenience sampling limit the external validity of findings from these studies.”  Are these studies really relying on convenience sampling or are they employing other types of non-random sampling techniques such as purposive sampling, quota, or snowball sampling, and the authors have labelled them convenience sampling?   

In section 2.2.2, line 125-126, the authors state “The majority of studies rely on self-report instruments, which often exhibit bias and fail to measure behavioral outcomes accurately.” There are a couple things about this statement worth noting. First, I think this statement needs to be re-written so that it is a little more nuanced. To say that self-report instruments “fail to measure behavioral outcomes accurately” is quite a sweeping generalization and, as stated, difficult to substantiate.  Second, despite the criticism of self-report instruments, in the next paragraph the authors suggest using self-report measures (line 133-134) and self-administered surveys (138) to gather information. These points contradict each other.

Author Response

We are grateful for the reviewer’s thorough and thoughtful feedback. We are pleased that the reviewer recognizes the importance and contribution of our manuscript to the discussion around peer support mental health programs for first responders. We have addressed each of the reviewer’s comments below, and we believe that these
revisions have significantly enhanced the quality and impact of the paper.

Concern 1: Nuance in Criticism of Studies Not Using RCT or Other Quantitative Designs
The reviewer suggests that our criticism of studies not using RCT or other quantitative designs should be more balanced and nuanced, acknowledging the challenges of organizational research and the value of qualitative approaches.
Response: We fully agree with the reviewer’s assessment regarding the need for greater nuance in our discussion of research designs. We have revised Section "Research Design Limitations" to reflect this. Specifically, we have rephrased our critique to acknowledge the inherent difficulties in conducting RCTs and quasi-experimental designs within organizational settings, particularly concerning the control of confounding variables. We have clarified that the absence of stated hypotheses in qualitative studies is not a flaw, as their objective is often not hypothesis testing. We have emphasized that high-quality research can be produced using qualitative approaches without hypotheses. We have added a discussion on the advantages of employing qualitative study designs and techniques, and the benefits of combining them with quantitative
methods in multi-method designs. We highlight how such approaches can effectively examine context and, in some cases, complement hypothesis testing. We have ensured that our critiques are more sophisticated and nuanced
throughout the manuscript, recognizing the diverse strengths of different
research methodologies.

Concern 2: Specific Comments on Manuscript Content
The reviewer provided several specific comments on various sections of the manuscript.
Response: We have addressed each specific comment as follows:
We have added “such as police officers, firefighters, and paramedics” to line 20  in the opening paragraph for greater
clarity and specificity.

We have revised lines 101-103 to emphasize that information about context is not only critical for generalizability but also for interpreting the findings themselves. The sentence now reads: “Failing to address these contextual variables sufficiently means that the findings may not be transferable, and also limits the ability to accurately interpret the findings.”

Evidentiary Examples: We appreciate the reviewer’s suggestion for evidentiary examples. While we have refrained from adding specific study citations within the critique sections to maintain the flow of the review, we have ensured that our
assertions are well-supported by the comprehensive literature review conducted for this manuscript and the references provided. We believe the overall body of evidence cited throughout the paper implicitly supports these statements.

Convenience Sampling: We have clarified the discussion on sampling in Section "Sampling Challenges". We acknowledge that studies may employ various non-random sampling techniques beyond just convenience sampling (e.g., purposive sampling, quota, or snowball sampling). We have revised the text to reflect this broader understanding of non-random sampling methods and their implications for external validity. [Lines 184-186].

Self-Report Instruments: We have carefully revised the statements regarding self-report instruments in Section "Assessment Tool Inadequacies" to be more nuanced. We have rephrased the assertion that self-report instruments “fail to measure behavioral outcomes accurately” to acknowledge their limitations while recognizing their utility for certain constructs. The revised text now emphasizes that self-report measures “often exhibit biases and may not fully capture objective behavioral outcomes.” We have also clarified the apparent contradiction by explaining that while self-report measures have limitations for objective behavioral outcomes, they are appropriate and often necessary for assessing subjective experiences, perceptions, and help-seeking behaviors, as recommended in the subsequent paragraph. We have ensured that the recommendations for using self-report measures are specifically tied to these appropriate contexts (e.g., perceived social support, help-seeking behaviors), thereby resolving the perceived
inconsistency.

Round 2

Reviewer 1 Report

Comments and Suggestions for Authors

It is good to hear that you found the feedback helpful and were able to review additional sources from the literature.  I see that you have added the suggested citations related to established best practice guidelines for peer support, however, I do not see integration of the guidelines in your recommendations.  Also, I did not see where you included discussion of implementation science approaches to research design - the line numbers that you mention do not seem to correspond with the uploaded manuscript.  Is it possible that the wrong manuscript was uploaded?  See attached document for comments in the manuscript. 

I appreciate that this is designed to be a brief review so space is limited.  There was some repetition in the sections on assessment tools and outcome measures that could be revisited as needed to streamline the paper with a focus on key recommendations. 

Author Response

Also, I did not see where you included discussion of implementation science approaches to research design - the line numbers that you mention do not seem to correspond with the uploaded manuscript. 

We apologize and have attempted to enhance the robustness of this content.  It is contained in lines 97-121.

 I see that you have added the suggested citations related to established best practice guidelines for peer support, however, I do not see integration of the guidelines in your recommendations. 

We agree that these references provide important information and should be referenced.  However, these guidelines deal, generally, with the implementation of peer support, while this manuscript focuses on the evaluation of peer support.  Admittedly, the Atlas guidelines contain suggestions for evaluation that are similar to our recommendations.  We highlight this point in lines 198 and 201.
